# A Novel Approach to Dual Feature Selection of Atrial Fibrillation Based on HC-MFS

**DOI:** 10.3390/diagnostics14111145

**Published:** 2024-05-30

**Authors:** Hong Liu, Lifeng Lu, Honglin Xiong, Chongjun Fan, Lumin Fan, Ziqian Lin, Hongliu Zhang

**Affiliations:** 1Business School, University of Shanghai for Science and Technology, Shanghai 200093, China; liuh@sumhs.edu.cn (H.L.); zoe6771@hotmail.com (L.F.); hongliuzhang1@gmail.com (H.Z.); 2Chongming Hospital, Shanghai University of Medicine & Health Sciences, Shanghai 202150, China; 3Collaborative Innovation Center for Biomedicine, Shanghai University of Medicine & Health Sciences, Shanghai 201318, China; 4Antai College of Economics and Management, Shanghai Jiao Tong University, Shanghai 200030, China

**Keywords:** atrial fibrillation, machine learning, clustering, feature selection

## Abstract

This investigation sought to discern the risk factors for atrial fibrillation within Shanghai’s Chongming District, analyzing data from 678 patients treated at a tertiary hospital in Chongming District, Shanghai, from 2020 to 2023, collecting information on season, C-reactive protein, hypertension, platelets, and other relevant indicators. The researchers introduced a novel dual feature-selection methodology, combining hierarchical clustering with Fisher scores (HC-MFS), to benchmark against four established methods. Through the training of five classification models on a designated dataset, the most effective model was chosen for method performance evaluation, with validation confirmed by test set scores. Impressively, the HC-MFS approach achieved the highest accuracy and the lowest root mean square error in the classification model, at 0.9118 and 0.2970, respectively. This provides a higher performance compared to existing methods, thanks to the combination and interaction of the two methods, which improves the quality of the feature subset. The research identified seasonal changes that were strongly associated with atrial fibrillation (pr = 0.31, FS = 0.11, and DCFS = 0.33, ranked first in terms of correlation); LDL cholesterol, total cholesterol, C-reactive protein, and platelet count, which are associated with inflammatory response and coronary heart disease, also indirectly contribute to atrial fibrillation and are risk factors for AF. Conclusively, this study advocates that machine-learning models can significantly aid clinicians in diagnosing individuals predisposed to atrial fibrillation, which shows a strong correlation with both pathological and climatic elements, especially seasonal variations, in the Chongming District.

## 1. Introduction

Atrial fibrillation (AF) is the third most common cardiovascular disease after hyper-tension and coronary heart disease. Although AF is not immediately life-threatening, it can lead to embolic complications and stroke, making it a significant predisposing factor for cerebrovascular events. According to the Global Burden of Disease Study 2019, there are more than 38 million cases of AF worldwide, and the disability-adjusted life years due to AF is 121.62 per 100,000 people in our country [1]. This puts a huge strain on the global health system. In 2021, statistical studies revealed significant differences in the prevalence of AF up to 12-fold across geographic regions [2], with North America, Europe, Southeast Asia, and China being particularly affected. In China, as a region with rapid population growth, the burden of AF will increase [3]. The variation in AF prevalence is due to a variety of factors, such as age, diet, and other diseases, making AF an important challenge in the field of smart healthcare. However, machine learning has shown great promise in improving the prediction of diseases, including atrial fibrillation [4]. It has been proven to be effective in processing large-scale healthcare data and discovering underlying disease patterns, risk factors, and trends, thus contributing to improved rates of early diagnosis of disease, personalized treatment, and patient management [5].

AF is a cardiac arrhythmia that has a significant impact on global health. According to the Global Burden of Disease study [1], AF is a leading contributor to 369 diseases and injuries worldwide, based on data from 2019. This systematic analysis has robustly substantiated the significance of AF in the context of the Global Burden of Disease study, emphasizing its impact on global health. Following this, a multinational cohort study involving 153,152 middle-aged individuals [2] revealed significant variations in the global prevalence, treatment approaches, and health implications of AF. The study disclosed that individuals with AF demonstrated an annual stroke incidence of 1.1% and a mortality rate of 2.6%, which were five-fold and two-fold as high as those in the healthy population, respectively. These elevated rates of morbidity and mortality associated with AF, coupled with the current lack of high-quality therapies, have positioned AF as a focal point for clinical research.

Scholars are currently investigating the risk factors associated with AF. Hyperuricemia has been identified as an independent risk factor for AF, raising clinical concerns for patients with hyperuricemia [6]. The relationship between alcohol and AF has also garnered significant attention. Voskoboinik et al. [7] explored this correlation, shedding light on the potential impact of alcohol consumption on AF. Furthermore, in a study examining the connection between epicardial fat distribution and left heart size [8], Nattel S [9] emphasized the influence of cardiac anatomy on AF, contributing to a deeper understanding of its pathogenesis. Earlier research has revealed associations between AF and climatic factors, such as air pollution [10,11,12] and temperature [13]. In a large Danish population-based study, Murphy N F et al. [14] observed a significant seasonal correlation between temperature and the incidence of AF, particularly a higher occurrence during colder seasons [15], in hospitalized patients with acute ischemic stroke. Their data are compelling and strongly support the association between temperature and AF, opening a new avenue for investigating the role of climatic factors in AF development. However, it is important to note that existing studies predominantly focus on the correlation between specific factors and AF, such as total cholesterol and lipoproteins, which are strongly associated with cardiovascular disease, including atrial fibrillation (AF) [16,17]. With the emergence of the big data era, the extraction of key insights from vast amounts of high-dimensional data is crucial for disease prediction. In the context of AF, a notable research gap exists. Therefore, this paper aims to concentrate on the influence of biochemical indicators, such as platelets and uric acid, on AF, while considering the climatic characteristics of the Chongming District. The goal is to extract key features from high-dimensional data [18], providing a fresh perspective for a more in-depth understanding of AF pathogenesis. Additionally, this paper aims to conduct a comprehensive examination of the relationships between these factors.

In recent years, substantial strides have been achieved in the realm of machine learning applied to the medical domain, particularly in its pivotal role in disease prediction [19]. A notable area of focus in the current research on AF revolves around two key aspects: feature selection and AF prediction. Within the realm of feature selection, researchers predominantly concentrate on extracting features from electrocardiogram signals and intracardiac electrical tracings [20,21]. However, there exists a notable gap in the extraction of key features from high-dimensional biochemical metrics. While the filter remains a widely used feature-selection method in this field, it tends to overlook the intricate interrelationships between features. Optimized feature-selection methods, such as hierarchical clustering feature selection [22], mutual information and hierarchical clustering [23], mutual information feature selection, and max-relevance and min-redundancy, among others, offer an improvement by computing the correlation between each feature and the target variable, as well as the correlation among features. These methods comprehensively consider the balance between these correlations. Nevertheless, these approaches typically follow a static greedy search mechanism, where the selected features remain unchanged or are not later adjusted. This limitation hampers their ability to find a subset of features with optimal performance. To address this shortfall, Hancer E et al. introduced MIRFFS [24], which utilizes forward and backward search techniques to identify the optimal feature subset. Building upon this concept, this paper innovatively proposes a dual feature-selection method based on hierarchical clustering and mean Fisher scores. In this approach, features are classified into relevant and irrelevant classes through hierarchical clustering. The Fisher scores are then employed to compute intra- and inter-class distances, determining the importance of features [25]. The combination of these two methods filters a high-performance subset of features. In the experiments aimed at finding the optimal feature subset, forward search and backward search were introduced to mitigate the performance loss caused by the static greedy mechanisms in previous studies. Additionally, data preprocessing includes the standardization of datasets, as well as handling missing values and outliers. Comparative experiments were conducted on four datasets with or without normalization and outlier processing, using five classification models (K-nearest Neighbor (KNN), Random Forest (RF), Support Vector Machine (SVM), Naive Bayes (NB), and the Logistic Regression Model (LR)) to observe the impact of normalization and outlier processing on prediction performance in order to identify the best-performing datasets and feature subsets. The experimental results demonstrate that the HC-MFS enhances the classification performance of the model, revealing a strong correlation between AF and platelets and season.

The structure of this paper is as follows: Section 2 presents the sources of the experimental data and outlines the experimental methods; Section 3 describes the execution of the experiments and the acquisition of the results; Section 4 discusses the findings and offers pertinent recommendations; and Section 5 provides a summary and conclusion.

## 2. Data Sources and Methods

### 2.1. Data Sources

This paper presents a retrospective study utilizing the AF dataset acquired from January 2020 to September 2023 at the Cardiovascular Department of a tertiary hospital in Chongming District, Shanghai. Furthermore, climatic data for the Chongming District was gathered from both the Shanghai Meteorological Bureau” http://sh.cma.gov.cn (accessed on 14 September 2023)” and the Shanghai Bureau of Ecology and Environment “https://sthj.sh.gov.cn (accessed on 14 September 2023)”. A comprehensive set of 20 features was meticulously selected by recommendations from coronary heart disease experts at a tertiary hospital in Chongming District and insights derived from prior research. Notably, the dataset exhibited missing values, prompting the removal of entries where more than 5 feature values were absent for the same ID. Consequently, 678 records were extracted, encompassing 339 patients diagnosed with AF and an equal number of patients without AF. To enhance the dataset’s quality, various preprocessing steps were implemented, including handling missing values, identifying and addressing outliers, and standardization. Subsequent phases of this study involved comparative experiments to evaluate the classification performance of the dataset with and without standardization and outlier treatments.

### 2.2. Statistical Analysis

The data were analyzed using Python 3.9, the measurement information was expressed as ± s, the counting information was expressed as the number of cases (%), comparisons between the groups were made using the independent samples *t*-test or the χ^2^ test, and the difference was considered statistically significant at *p* < 0.05. CO, minimum temperature, CRP, platelets, platelet pressure product, LDL, HDL, and TC were lower in the atrial fibrillation group than in the non-atrial fibrillation group (control); age, PM_10_, NO_2_, erythrocyte distribution width, erythrocyte pressure product, platelet distribution width, large platelet ratio, mean platelet volume, and uric acid were higher than those in the control group; and the differences were all statistically significant (*p* < 0.05), as shown in Table 1.

### 2.3. Dual Feature-Selection Method Based on Hierarchical Clustering and Mean Fisher Score HC-MFS

The inclusion of weak or irrelevant features within disease data often compromises the performance of disease prediction models. Traditional feature-selection methods rigidly lock in selected features, hindering the identification of an optimal subset for improved model performance. To address these challenges, we present the HC-MFS method. The HC-MFS method employs hierarchical clustering to categorize features into classes, distinguishing irrelevant ones and reducing the similarity within each class. Simultaneously, Fisher scoring evaluates feature importance by computing intra- and inter-class distances, thereby selecting features most relevant to the target. The combination of these two methods, coupled with forward and backward search strategies, results in the identification of a feature subset with superior performance. The HC-MFS process is shown in Figure 1. The method unfolds in the following steps:

Step 1. Ward Clustering: Apply Ward clustering to the original feature sequence *F* = {*f*_1_, *f*_2_, …, *f_n_*}.

Step 2. Fisher Score Calculation: Compute Fisher scores (FS) for features, rank them, and determine each category’s mean Fisher score (MFS).

Step 3. Initial Feature Subset Selection: Include features with FS higher than MFS in the initial feature subset *F_FSS_*, emphasizing superior class differentiation ability.

Step 4. Iterative Refinement: Based on *F_FSS_*, iteratively enhance model accuracy by adding features through forward and backward searches until all features are considered.

Step 5. Accuracy Comparison: Compare the accuracy of the forward search (F) and backward search (R), selecting the subset with higher accuracy. In the case of a tie, prioritize the subset with fewer features.

## 3. Empirical Analysis

### 3.1. Parameter Settings

In this paper, we conducted experiments using Python 3.9 to compare HC-MFS with FS, Relief_F (R_F), mutual information (MI), and the Distance Correlation Factor (DCFS) on the same dataset using the KNN, RF, SVM, NB, and LR models. We aimed to find the optimal combinations of the model’s parameters using lattice scavenging and 5-fold cross-validation. The experimental results show that datasets with missing-value processing and standardization (IMV + SS) reach optimal performance more frequently, indicating that most datasets benefit from standardization treatment in improving model performance. At the same time, some models require outlier treatment (OR) to further optimize performance. However, the use of mean substitution for outliers in this paper was found not to achieve higher performance in RF models. Complex methods such as median substitution, truncation, and distribution-based methods can be considered to better preserve the original data characteristics. Abbreviations in the text are explained in Table 2. The specific parameter settings of the model are shown in Table 3, and Further details experiments are shown in Table A1 in Appendix A.

### 3.2. Clustering Results

The results of Ward’s method of clustering are presented in Figure 2 The names of the features are displayed at the lower end of the clustering dendrogram. The results indicate that it is more appropriate to classify the features into three categories. The first major class is dominated by the seasonal and cholesterol classes, the second class is dominated by the age and red blood cell classes, and platelets are a separate class. The increased gap between classes allows subsequent feature selection to reduce the correlation between features and improve the quality of the feature subset. The separate category of platelets, on the other hand, provides a new direction for our study, which is analyzed in detail in the subsequent Section 4.1.3.

### 3.3. Model Evaluation and Comparison

This paper investigates the correlation between all features and atrial fibrillation using different methods and evaluates their performance using a classification model. The training set achieved optimal performance in the backward search of the RF model by the HC-MFS method, with the corresponding feature subset containing 12 features. To maintain the same feature dimensions, we ranked the features according to the FS, R_F, MI, and DCFS methods, and selected the top 12 features as the feature subset for comparison experiments with HC-MFS. The mean correlation and initial feature dimensions of HC-MFS are given in Table 4, and the final feature subsets of the five methods are given in Table 5. The experimental results show that all five methods achieve optimal performance in the RF model, with accuracies of 0.7868, 0.7574, 0.7206, 0.7574, and 0.7647, respectively. HC-MFS performs better, as demonstrated in Table 6. The accuracy of HC-MFS was improved by up to 9.19%, and the root mean square error was reduced by up to 9.69%, compared with the existing methods, details are shown in Table 7. These results indicate that the HC-MFS method plays a significant role in enhancing the performance of the model for predicting atrial fibrillation diseases. Moreover, a recent study by Andrea Bernardini et al. [26] on the prediction of atrial fibrillation based on machine learning showed that an AUC of 0.7790 ± 0.0160 was obtained, and our experiment is superior to this result, which is attributed to the introduction of hierarchical clustering to reduce feature redundancy.

## 4. Discussion and Suggestions

### 4.1. Feature Analysis

#### 4.1.1. Season Correlation Analysis

A significant association between season and AF was observed, with the DCFS value reaching 0.3298, the FS being 0.1043, and the Relief_F score 180 being 0.0603, all of which were at the top of the characteristic correlations. It is worth emphasizing that the Chongming District is located in the subtropical region, and its winter and summer last for more than 230 days, with an average annual number of rainy days as high as 93 to 183 days, and the annual rainfall is on an increasing trend, which is nearly twice as much as the national average annual rainfall. This unique climatic feature leads to elevated air humidity, which may aggravate the burden on the respiratory and circulatory systems, providing a potential physiological explanation for the onset of AF. In addition, the continuous decrease in the average annual minimum temperature in the Chongming District over the past three years, from 15 degrees to 13 degrees Celsius, has been more pronounced, especially in the spring and winter of 2022. This decreasing temperature trend may be associated with the increasing incidence of AF, providing a substantial observational basis for a further in-depth investigation of the relationship between climatic factors and AF. Therefore, we suggest that the unique climatic conditions and the decreasing temperature trend in the Chongming District may be important influences on the incidence of AF.

#### 4.1.2. Serum Total Cholesterol (TC) and Low-Density Lipoprotein Correlation Analysis

Metabolic syndrome is widely recognized as an independent risk factor for AF, which mainly encompasses hypertension, diabetes mellitus, and dyslipidemia [27,28]. A 5-year follow-up study [29] demonstrated that subjects with incident AF were typically older, had higher rates of obesity, and more severe alcohol abuse, and their baseline BMI, TC, and glucose levels were higher compared with those without AF. In particular, the high variability in TC levels resulted in a 7.5% increased risk of AF. The reasons for this were analyzed: First, high levels of TC and LDL are strongly associated with the development of atherosclerosis. Hardened arteries lead to the obstruction of blood flow and increase the burden on the heart. Second, high levels of TC and LDL may trigger inflammatory responses and oxidative stress, and these inflammatory processes may adversely affect cardiac electrophysiology and atrial tissue. Again, high levels of TC and LDL may lead to myocardial fibrosis, abnormal proliferation, and the hardening of cardiac muscle tissue, which affects electrical signaling in the atria. In addition, high cholesterol may lead to alterations in platelet aggregation and the clotting process, increasing the risk of blood clots, which can travel through the bloodstream to the atria. Finally, persistently high levels of cholesterol and LDL can lead to the enlargement of the atria and heart failure. These pathological conditions increase the pressure and volume load on the left atrium, leading to atrial fibrosis and electrical conduction abnormalities, thereby increasing the risk of atrial fibrillation.

#### 4.1.3. Platelet Count Correlation Analysis

The experimental results presented a striking finding of a positive correlation between platelets and AF, a conclusion that provides an answer to a question that previous studies have failed to clarify. Specifically, the mutual information values were 0.296 for P-LCR and AF and 0.2627 for PLT, which were at the forefront in terms of characteristic correlation. We used PLT as a sole indicator to predict AF. The experimental results showed that the accuracy of predicting AF using only platelet count was over 60%, with a recall rate of 64.29%. This indicates that platelet count has a certain role in predicting AF. Although the predictive capability of PLT as a single indicator is limited, its performance still demonstrates some clinical value. By combining PLT with other biomarkers and clinical characteristics, the accuracy and reliability of AF prediction could be further improved. The reasons for this were analyzed: First, patients with AF are usually accompanied by AF and irregular heartbeats, which can increase the risk of platelet involvement in thrombosis. AF may lead to the retention of blood in the atria and the formation of blood clots, which directly increases the risk of embolism. Second, platelets play a key role in the pathophysiological process of inflammation and vascular endothelial injury, which is one of the key factors in the development of AF. In this process, platelets contribute to the inflammatory response and repair of endothelial damage through adhesion and the release of inflammatory mediators, thereby exacerbating the progression of cardiovascular disease. In addition, abnormal platelet activity and function may lead to the formation of microthrombi in blood vessels. The presence of these microthrombi may cause microcirculatory disorders in the atria, which in turn promote the development of AF. The above analysis provides a more comprehensive understanding of the association between platelets and AF and provides useful clues for an in-depth exploration of the pathogenesis of AF.

#### 4.1.4. C-Reactive Protein (CRP) Correlation Analysis

Our study found that CRP was positively correlated with AF, with a high mutual information value of 0.5809, which ranked first in the ranking of characteristics, a result consistent with previous studies. The pathogenesis of AF is complex and involves the interaction of inflammatory responses and atrial fibrosis. These processes lead to the electrical and structural remodeling of the atria [30]. There is growing evidence that chronic inflammation can alter the electrophysiology of the atria, driving myocardial fibrosis and structural changes that affect the electrophysiological properties of the heart and may increase susceptibility to AF. In addition, elevated CRP levels are associated with thrombosis and vascular endothelial dysfunction. CRP can directly impair endothelial function, leading to a pro-thrombotic state and increasing the risk of thromboembolic events in patients with AF. Endothelial dysfunction also contributes to atrial remodeling and AF progression. In AF, irregular atrial contractions lead to blood retention in the atria, increasing the risk of thrombosis. High CRP levels may indirectly increase the likelihood of thrombosis by causing vascular endothelial damage and inflammatory responses.

### 4.2. Suggestions

#### 4.2.1. Suggestions to Patients

To reduce the risk of AF, patients are advised to follow these guidelines:

(1) Stay warm in cold weather by wearing adequate clothing to reduce the risk of hypothermia and minimize outdoor activity.

(2) Adopt a heart-healthy diet that includes reducing saturated fat and cholesterol intake and increasing the intake of fruits and vegetables, grains, fish, and healthy fats (e.g., nuts), as well as quitting smoking and limiting alcohol intake.

(3) Exercise for at least 150 min of moderate-intensity aerobic exercise per week, such as brisk walking and cycling.

(4) Undergo regular health checkups and cardiac screening for early detection and the management of risk factors if you are in a group at high risk of AF.

#### 4.2.2. Suggestions for Hospitals

To improve the prevention and treatment of AF, the following guidelines are recommended:

(1) Provide health promotion education on the potential risks to heart health during the cold season, as well as preventive measures.

(2) Increase medical resources by ensuring that hospitals have adequate equipment and staff to meet patient needs during the winter months and that they have a dedicated cardiac emergency room.

(3) Prevent the spread of infection by providing influenza vaccines, improving hand hygiene, and taking isolation measures during the winter months when influenza and other respiratory infections are more likely to spread.

(4) Regularly screen patients for CRP, TC, PLT, and other biochemical indicators, especially during physical exams and outpatient clinics, to focus on high-risk groups.

(5) Establish a multidisciplinary team of cardiovascular specialists, internists, dietitians, and psychologists to actively research and innovate to continuously improve the prevention and treatment of AF. Enhance cooperation and coordination with other healthcare providers, insurance companies, government agencies, and non-profit organizations to provide more comprehensive AF management and care.

## 5. Conclusions

In this study, we developed a novel dual feature-selection method for atrial fibrillation (AF) by combining hierarchical clustering and Fisher score methods, termed HC-MFS. We used patient data from a tertiary hospital in Chongming District, Shanghai, and climate data from the Chongming District provided by the Shanghai Meteorological Bureau and the Shanghai Bureau of Ecology and Environment. We validated HC-MFS with empirical examples and compared its performance with established methods such as feature selection, Relief_F, mutual information, and DCFS. Our results show that HC-MFS outperforms these methods, achieving the lowest root mean square error. Key features identified include season, total cholesterol, low-density lipoprotein, platelet count, and C-reactive protein. This study offers valuable insights for disease prediction and serves as a reference for physicians during diagnosis.

## 6. Limitations

Several important limitations of this study must be emphasized. The amount of data obtained for this study was limited, and there were some limitations in model training. Moreover, hospital inpatient data were used for all data; outpatient diagnoses were not included in our analysis and may have been underestimated. Echocardiographic, biological, and imaging parameters were also not available. In other words, to improve the model quality, there is a need for more data. We also encourage other researchers to explore these directions for AF development.

## Figures and Tables

**Figure 1 diagnostics-14-01145-f001:**
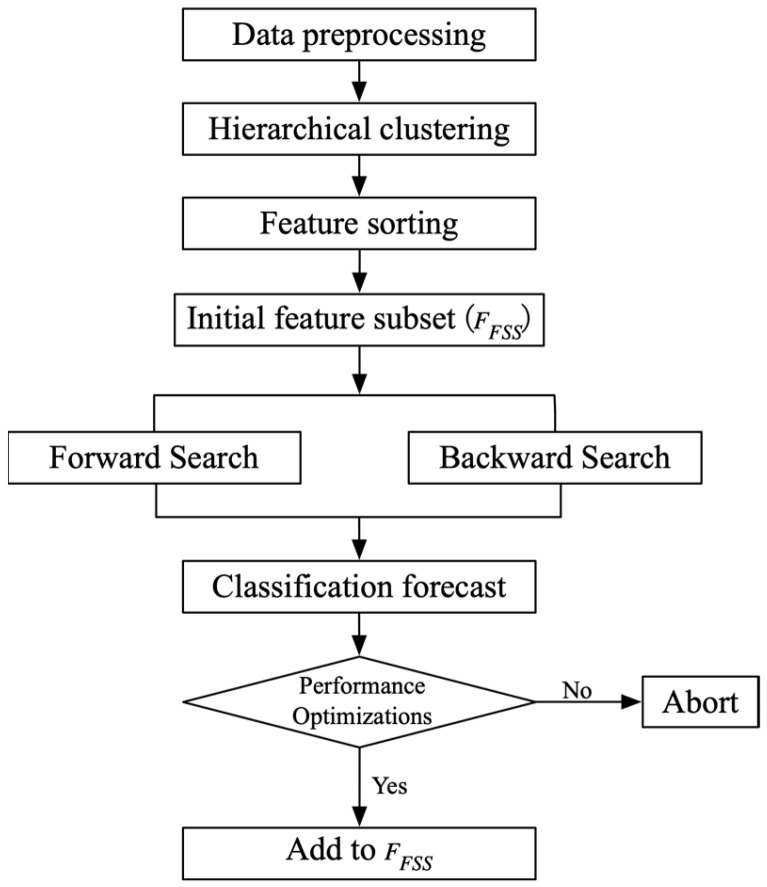
Flowchart of HC-MFS method.

**Figure 2 diagnostics-14-01145-f002:**
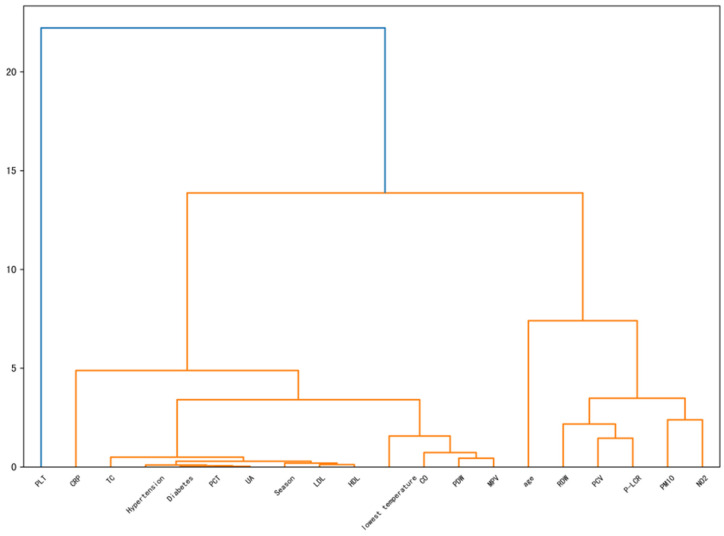
Hierarchical clustering results. Because platelets (PLT) are a unique category and quite special, they are highlighted in blue. The orange color has no special significance.

**Table 1 diagnostics-14-01145-t001:** Basic overview of AF and statistical analysis.

Features	Non-AF Group (*n* = 339)	AF Group (*n* = 339)	t/χ^2^	*p*-Value
Age/years	76.53 ± 11.16	79.48 ± 9.45	−3.717	<0.001
CO/ppm	14.94 ± 4.44	13.82 ± 4.76	3.187	0.002
Minimum Temperature/°C	13.87 ± 9.45	11.78 ± 9.49	2.874	0.004
Seasons			36.091	<0.001
Spring/n	68 (20.1%)	112 (33.0%)		
Summer/n	65 (19.2%)	99 (29.2%)		
Autumn/n	95 (28.0%)	61 (18.0%)		
Winter/n	111 (32.7%)	67 (19.8%)		
CRP/(mg·L^−1^)	27.74 ± 55.00	9.52 ± 12.03	2.950	0.003
Platelets/(L^−1^)	194.91 ± 76.04	169.49 ± 56.80	4.168	<0.001
Platelet Distribution Width/(%)	12.43 ± 2.37	13.31 ± 2.88	−4.328	<0.001
Large Platelet Ratio/(%)	29.90 ± 8.33	31.86 ± 8.11	−3.709	<0.001
Platelet Crit/(%)	0.21 ± 0.08	0.19 ± 0.69	3.121	0.0019
Mean Platelet Volume/(fL)	10.63 ± 1.09	10.91 ± 1.11	−4.009	<0.001
LDL/(mmol·L^−1^)	2.38 ± 0.99	2.04 ± 0.79	4.600	<0.001
Uric Acid/(μmol·L^−1^)	0.34 ± 0.13	0.38 ± 0.13	−5.352	<0.001
TC/(mmol·L^−1^)	4.38 ± 1.26	3.90 ± 1.03	4.534	<0.001
PM_10_/(μg/m^3^)	31.97 ± 16.75	34.99 ± 17.35	−2.306	0.021
NO_2_/(ppbv)	22.14 ± 12.54	24.24 ± 14.69	−2.002	0.046
Diabetes/n			2.446	0.015
Diabetes/n	129 (38.1%)	99 (29.2%)		
Not Diabetes/nic	210 (61.9%)	240 (70.8%)		
Hypertension/n			−2.664	0.024
Hypertension/n	236 (69.6%)	262 (77.3%)		
Not Hypertension/n	103 (30.4%)	77 (22.7%)		
Erythrocyte Distribution Width/(fL)	45.13 ± 6.12	46.14 ± 6.46	−2.072	0.039
Erythrocyte Pressure/(L/L)	37.38 ± 6.92	38.37 ± 6.30	−1.959	0.049
High-Density Lipoprotein (HDL)	1.61 ± 0.65	1.51 ± 0.46	2.168	0.031

**Table 2 diagnostics-14-01145-t002:** Abbreviation explanation.

Abbreviation	Explanation
HC-MFS	A novel dual feature-selection methodology, combining hierarchical clustering with Fisher scores
FS	Fisher score feature-selection method
R_F	Relief_F feature-selection method
MI	Mutual information feature-selection method
DCFS	Distance Correlation Factor feature-selection method
KNN	A classification model: K-nearest Neighbor
RF	A classification model: Random Forest
SVM	A classification model: Support Vector Machine
NB	A classification model: Naive Bayes
LR	A classification model: the Logistic Regression Model
IMV	Only missing values are processed on the dataset.
IMV+SS	The dataset is processed for missing values and standardized.
IMV+OR	The dataset is processed for missing values and outliers.
IMV+OR+SS	The dataset is processed for missing values, outlier values, and standardization.

**Table 3 diagnostics-14-01145-t003:** The parameter settings for the five categorical prediction models.

Methods	Model	Parameter	Parameter Value	Datasets
HC-MFS	KNN	n_neighbors	19	IMV+OR+SS
		p	1	
		weights	distance	
	RF	max_depth	16	IMV+SS
		random_state	2	
		n_estimators	300	
	SVM	C	1	IMV
		kernel	linear	
	NB	/	GaussianNB	IMV+SS
	LR	solver	liblinear	IMV
		penalty	L1	
		C	10	
FS	KNN	n_neighbors	30	IMV+SS
		p	/	
		weights	uniform	
	RF	max_depth	5	IMV+SS
		random_state	2	
		n_estimators	300	
	SVM	C	1	IMV+SS
		kernel	linear	
	NB	/	GaussianNB	IMV
	LR	solver	liblinear	IMV+SS
		penalty	L2	
		C	1	
R_F	KNN	n_neighbors	18	IMV+SS
		p	1	
		weights	distance	
	RF	max_depth	3	IMV+SS
		random_state	1	
		n_estimators	100	
	SVM	C	1	IMV+OR+SS
		kernel	linear	
	NB	/	GaussianNB	IMV+OR+SS
	LR	solver	liblinear	IMV
		penalty	L1	
		C	10	
MI	KNN	n_neighbors	20	IMV+SS
		p	1	
		weights	distance	
	RF	max_depth	19	IMV+SS
		random_state	1	
		n_estimators	200	
	SVM	C	1	IMV
		kernel	linear	
	NB	/	GaussianNB	IMV
	LR	solver	liblinear	IMV+SS
		penalty	L2	
		C	0.1	
DCFS	KNN	n_neighbors	28	IMV+SS
		p	/	
		weights	uniform	
	RF	max_depth	4	IMV+OR+SS
		random_state	2	
		n_estimators	100	
	SVM	C	1	IMV+SS
		kernel	linear	
	NB	/	GaussianNB	IMV+SS
	LR	solver	liblinear	IMV+SS
		penalty	L2	
		C	1	

**Table 4 diagnostics-14-01145-t004:** Initial feature subset selection for HC-MFS.

Method	Classification	Mean	Initial Feature Subsets
HC-MFS	1	0.0260	8
	2	0.0111	
	3	0.0257	

**Table 5 diagnostics-14-01145-t005:** Feature subsets.

Method	Feature Subsets
HC-MFS	{Season, uric acid, low-density lipoprotein, total cholesterol, platelet distribution width, platelets, mean platelet volume, age, large platelet ratio, C-reactive protein, high-density lipoprotein, and erythrocyte distribution width}
FS	{Season, uric acid, LDL, total cholesterol, platelet distribution width, platelets, mean platelet volume, age, large platelet ratio, CO, platelet pressure, and C-reactive protein}
R_F	{Season, hypertension, large platelet ratio, diabetes mellitus, high-density lipoprotein, erythrocyte pressure product, C-reactive protein, total cholesterol, low-density lipoprotein, platelet pressure product, platelets, and age}
MI	{C-reactive protein, total cholesterol, LDL, large platelet ratio, platelets, HDL, erythrocyte pressure volume, erythrocyte distribution width, season, platelet distribution width, uric acid, and PM10}
DCFS	{Season, uric acid, total cholesterol, LDL, platelet distribution width, age, platelets, mean platelet volume, large platelet ratio, CO, HDL, and C-reactive protein}

**Table 6 diagnostics-14-01145-t006:** Model results.

Method	Accuracy	Precision	Recall	F1	AUC	Datasets
HC-MFS	0.7868	0.7887	0.8000	0.7943	0.7964	IMV+SS
FS	0.7574	0.7403	0.8183	0.7755	0.7556	IMV+SS
R_F	0.7206	0.7162	0.7571	0.7361	0.7195	IMV+SS
MI	0.7574	0.7467	0.8000	0.7724	0.7561	IMV+SS
DCFS	0.7647	0.7639	0.7857	0.7746	0.7641	IMV+OR+SS

**Table 7 diagnostics-14-01145-t007:** Model errors.

Method	RMSE	R	std
HC-MFS	0.4774	0.5437	0.4769
FS	0.4926	0.5156	0.4899
R_F	0.5286	0.4405	0.5278
MI	0.4926	0.5146	0.4912
DCFS	0.4851	0.5288	0.4848

## Data Availability

Due to the sensitive nature of the data used in this study, which include personal and confidential patient information, the datasets are not publicly available. Data sharing is restricted to safeguard the privacy and confidentiality of the participants as per applicable laws and regulations. Detailed data access policies and requests for collaboration may be directed to the corresponding author, subject to approval by the relevant ethics and data protection authorities.

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
