# Peer review of "A Novel Approach to Dual Feature Selection of Atrial Fibrillation Based on HC-MFS"

_diagnostics, 2024, doi:10.3390/diagnostics14111145_

Round 1
Reviewer 1 Report
Comments and Suggestions for Authors
The following points need to be addressed in the revision:
- The abstract is adequate in length and structure.
- Mention the public availability of the designated dataset mentioned in the abstract.
- The highest accuracy mentioned in the abstract is not clear. What it is related to? Which it is compared with? Clear this in the abstract.
- Thoroughly go through your article for typos / grammatical mistakes.
- A single system diagram of the proposed solution is missing in the article.
- The first paragraph (lines 18-34) in the Introduction Section needs multiple citations which are missing right now. Correct this problem.
- The main innovation and contribution of this research should be clarified in the introduction.
- Lines 57-81 have no citations from the references. Cite the work of others.
- The literature survey is weak for this article. I would like to ask the authors to make a recent survey of articles related to this study and rewrite this portion of the article.
- Lines 96-100 highlight the work in all. It is not sufficient for a well-reputed journal article. Add a list of contributions of your work represented in this study at the end of the introduction.
- Add text for the organization of the article.
- Section 2.1 should share the link of its public availability for checking the reproducibility of your results.
- Make the caption of Table 1 more elaborative.
- Table 1 has not been used in the text before its appearance. Most of the terms in Table 1 need an introduction in the text.
- Remove the sentence like “No further experiments will be conducted in this paper”. How the specific parameters setting was achieved? Add details with a flow chart in the article.
- In Section 3.2, giving the resulting diagram is not sufficient. Why not have a discussion added here about the results? The diagram is not self-explanatory.
- The analysis part of the article is quite weak. It needs improvement.
- Why the test results of your work are better than training results? It is an unusual scenario. It can be attributed to the potential model instability. One of the many reasons can be the use of a smaller dataset with repeated instances. You have used SMOTE to deal with data instability, but the results without this are missing. Also, check the data leakage problem that can lead to this problem.
- The statistical analysis of this work is missing and needs to be provided.
- Lines 178-240 are not appropriately suited to the article. Try moving them somewhere relevant portion. Why these lines are without any citations?
- A comparison with recent studies is missing in the article.
- Section 4.2 is again out of context without any proper linkages to the work in the article.
- I am not satisfied with the claims given in the conclusion. I suggest rephrasing a summarized conclusion in line with the article without any unrelated text.
Comments on the Quality of English Language
Moderate editing would do the job.
Author Response
Dear Reviewer,
We would like to take this opportunity to thank you for your precious time and great efforts in handling the review process of our previous submission. We have thought over all the comments and made our best effort to revise the paper; accordingly, we have also carefully polished the paper to improve the presentation. All changes were highlighted, with red color and strikethrough font for deletions and blue for additions. We hope that the revised version satisfies the requirement for publication in Diagnostics.
Enclosed please find the point-by-point responses to the reviewers’ comments. Please kindly let us know if there are any additional comments/concerns. We are looking forward to hearing from you.
Best regards,
Honglin Xiong

Reviewer 2 Report
Comments and Suggestions for Authors
The manuscript is devoted to the early identification of risk factors of atrial fibrillation using hierarchical clustering with Fisher scores. The authors are analyzed some characteristics including season, uric acid, low-density lipoprotein, total cholesterol, platelet distribution width, platelets, mean platelet volume, age, large platelet ratio, C-reactive protein, high-density lipoprotein, erythrocyte distribution width.
The authors concluded that machine learning models can significantly help the clinicians in diagnosing individuals predisposed to atrial fibrillation and suggested practical recommendations concerning a balanced diet, regular exercise, maintaining warmth for patients and educational efforts.
Minor comment: the shortening of the introduction can improve the perception of the article.
Author Response

(The authors gave the same response as above.)

Reviewer 3 Report
Comments and Suggestions for Authorsсomment in app

Author Response

(The authors gave the same response as above.)

Round 2
Reviewer 1 Report
Comments and Suggestions for Authors
All my points have been thoroughly addressed. The article is in good shape now.